# Color-Dense Illumination Adjustment Network for Removing Haze and Smoke from Fire Scenario Images

**DOI:** 10.3390/s22030911

**Published:** 2022-01-25

**Authors:** Chuansheng Wang, Jinxing Hu, Xiaowei Luo, Mei-Po Kwan, Weihua Chen, Hao Wang

**Affiliations:** 1State Key Laboratory of Nuclear Power Safety Monitoring Technology and Equipment, China Nuclear Power Engineering Co., Ltd., Shenzhen 518172, China; wangcs95@163.com (C.W.); chenweihua@cgnpc.com.cn (W.C.); wang_h@cgnpc.com.cn (H.W.); 2Shenzhen Institute of Advanced Technology, Chinese Academy of Sciences, Shenzhen 518055, China; 3Department of Architecture and Civil Engineering, City University of Hong Kong, Kowloon, Hong Kong 999077, China; xiaowluo@cityu.edu.hk; 4Institute of Space and Earth Information Science, The Chinese University of Hong Kong, Shatin, Hong Kong 999077, China; mpk654@gmail.com

**Keywords:** haze removal, visual enhancement, aerosol scattering model

## Abstract

The atmospheric particles and aerosols from burning usually cause visual artifacts in single images captured from fire scenarios. Most existing haze removal methods exploit the atmospheric scattering model (ASM) for visual enhancement, which inevitably leads to inaccurate estimation of the atmosphere light and transmission matrix of the smoky and hazy inputs. To solve these problems, we present a novel color-dense illumination adjustment network (CIANet) for joint recovery of transmission matrix, illumination intensity, and the dominant color of aerosols from a single image. Meanwhile, to improve the visual effects of the recovered images, the proposed CIANet jointly optimizes the transmission map, atmospheric optical value, the color of aerosol, and a preliminary recovered scene. Furthermore, we designed a reformulated ASM, called the aerosol scattering model (ESM), to smooth out the enhancement results while keeping the visual effects and the semantic information of different objects. Experimental results on both the proposed RFSIE and NTIRE’20 demonstrate our superior performance favorably against state-of-the-art dehazing methods regarding PSNR, SSIM and subjective visual quality. Furthermore, when concatenating CIANet with Faster R-CNN, we witness an improvement of the objection performance with a large margin.

## 1. Introduction

The phenomenon of images degradation from fire scenarios is usually caused by the large number of suspended particles generated during combustion. When executing the robot rescue in such scenes, the quality of the images collected from the fire scenarios will be seriously affected [1]. For example, most of the current research in the computer vision community is based on the assumption that the input datasets are clear images or videos. However, burning is usually accompanied by uneven light and smoke, reducing the scene’s visibility and failing many high-level vision algorithms [2,3]. Therefore, removing haze and smoke from fire scenario scenes is very important to improve the detection performance for rescue robots and monitoring equipment.

Generally, the brightness distribution in the fire scenarios is uneven, and different kinds of materials will produce different colors of smoke when burning [4]. Therefore, the degradation of the images in fire scenarios is more variable than common hazy scenes. Optically, poor visibility in fire scenarios is due to the substantial presence of solid and aerosol particles of significant size and distribution in the participating medium [5,6]. Light from the illuminant reflected tends to be absorbed and scattered by those particles, causing degraded visibility of a scene. The brightness is not evenly distributed in the background. The uncertain light source is the two main factors that cause the hazy fire scenarios to be far different from the common hazy scenarios.

Recently, many dehazing algorithms for single images have been proposed [7,8,9,10], aimed at improving the quality of the images captured from hazy or foggy weather. Image dehazing algorithms can be used as a preprocessing step for many high-level computer vision tasks, such as video coding [11,12,13], image compression [14,15] and object detection [16], etc. The dehazing algorithms can be roughly divided into two categories: the traditional prior-based methods and the modern learning-based methods [17]. The conventional techniques get plausible dehazing results by designing some hand-crafted priors, which lead to color distortion due to lack of consideration and comprehensive understanding of the imaging mechanism of hazy scenarios [18,19,20]. Therefore, traditional prior-based dehazing methods are difficult to achieve desirable dehazing effects.

Learning-based dehazing methods adopt convolution neural networks (CNNs) to simulate the mapping relationships between the hazy images and the clear images [21]. However, since the parameters and weights of the model are fixed after training, the datasets will seriously affect the performance of learning-based dehazing methods. Therefore, the learning-based dehazing methods lack sufficient flexibility to deal with the changeable fire environment. In addition, many synthesized training datasets for the dehazing algorithms are based on the atmospheric scattering model (ASM) [22,23], in which only white haze can be synthesized, and the other potential colors of smoke cannot be synthesized. These limitations will affect the application of current leaning-based dehazing models in fire monitoring systems [24]. Therefore, both prior-based methods and the learning-based methods are limited in the fire scenario dehazing.

This paper modifies ASM and proposes a new imaging model named aerosol scattering model (ESM) for enhancing the quality of images or videos captured from fire scenarios. In addition, this paper also presents a novel deep learning model for fire scenarios image dehazing. Specifically, instead of directly learning an image-to-image mapping function, we design a three-branch network to handle the transmission, suspend the particle color, and obtain a preliminary dehazing result separately.

This strategy is based on two observations. Firstly, the illumination intensity of most fire scenarios is uneven, the area near the fire source is significantly brighter than other areas. Second, different types of combustion usually produce various forms of smoke. For example, solid combustion usually produces white smoke, while combustible liquids generally generate black smoke. Therefore, the degraded images of fire scenarios present generally different styles. The reliability of most current dehazing methods is higher only under even illumination conditions [25]. To address the above-mentioned problems, this paper proposed a novel CIANet that can effectively improve the haze images captured from fire scenarios.

The proposed method can be seen as a compensation process that can enhance the quality of the images affected by combustion. The network learns the features of the images from the training data. The three branches of the structure generate an intermediate result, a transmission map and a color value, respectively. To a certain extent, our method integrates all the conditions for compensating or repairing the loss caused by scattering. The improved ESM is post-processing to transform the intermediate results to higher-quality images. After ESM processes the intermediate results, the color of the image is brighter, and the contrast is more elevated. ESM can also be employed in conventional image dehazing tasks, especially natural conditions.

The contributions of this work are summarized as follows:This paper proposes a novel learning-based dehazing model to improve the quality of images captured from fire scenarios, built with CNN and a physical imaging model. Combining the modern learning-based strategy with a traditional ASM makes the proposed model handle various hazy images in the fire scenarios without incurring additional parameters and computational burden.To improve the effect of image dehazing, we improve the existing ASM and propose a new ASM called the aerosol scattering model (ESM). The ESM uses brightness, color, and the transmission information of the images and can generate a more realistic images without causing over enhancement.We conducted extensive experiments on multiple datasets, and experiments show that the proposed CIANet achieves better performance quantitatively and qualitatively. The detailed analysis and experiments show the limitation of the classical dehazing algorithms in fire scenarios. Moreover, the insights from the experimental results confirm what is useful in more complex scenarios and suggest new research directions in image enhancement and image dehazing.

The remaining part of this paper is organized as follows. In Section 1, we review the ASM and some state-of-the-art image dehazing algorithms. In Section 2, we present the proposed CIANet in detail. Experiments are presented in Section 3, before conclusion is drawn in Section 4.

## 2. Related Works

Generally, the existing image haze removal methods can be roughly divided into two categories: the prior-based and learning-based methods. The prior-based strategy use hand-crafted prior inspired by statistical assumptions, while the learning-based methods automatically obtain the nonlinear mapping between images pairs from the training data [26]. We will discuss the differences between the two paradigms in this chapter.

### 2.1. Atmospheric Scattering Model

The prior-based dehazing algorithms can be regarded as an ill-posed problem. In this line of methods, the physical imaging model and various prior statistics are used to estimate the expected results from degraded inputs. In the dehazing community, the most authoritative model is the ASM proposed by McCartney [27], which can be formulated as:(1)I(x)=J(x)t(x)+a(1−t(x))
where, J(x) is the clear images to be recovered, I(x) is the captured hazy images, α is the global atmospheric light, and t(x) is the transmission map. Equation (Equation 4) suggests that the clear images J(x) can be recovered after t(x) and α are estimated.

The transmission map t(x) describes that the light reaches the camera instead of being scattered and absorbed, t(x) is defined as:(2)$$t\left(x\right)=e^{-\beta d(x)}$$
where d(x) is the distance between the scene point and the imaging devices, and β is the scattering coefficient of the atmosphere. Equation (Equation 2) shows that t(x) approaches 0 as d(x) approaches infinity.

### 2.2. Prior-Based Methods

The unknown items α and t(x) are the main factors that cause the image dehazing problem to be ill-posed. Therefore, various traditional prior-based methods [28] have been proposed to obtain an approximate dehazed result. In [29], the authors adopted haze-lines prior for estimating the transmission. In [30], the transmission map is calculated by proposing a color attenuation prior, which exploited the attenuation difference among the RGB channels. The paradigm of these kind of methods are illustrated in Figure 1a.

However, these prior-based methods often achieve sub-optimal restoration performance. For example, He et al. [31] utilize the dark channel prior to estimate the transmission map t(x) and employ a simple method to estimate the atmospheric light value *a* for restoring the clear images according to Equation (Equation 1). However, the sky region in the hazy images suffers from a negative visual effect when a dark channel prior is used. Zhu et al. [30] propose the color attenuation prior (CAP) for estimating the depth information of the hazy images to estimate the transmission maps t(x). Berman et al. [29] propose the non-local prior for estimating the transmission maps t(x) of hazy images in RGB space by varying the color distances. Even though these prior-based methods can restore clear images from the hazy images, the process can easily lead to incorrect estimation of the atmospheric lights *a* and transmission map t(x), which cause the color distortion in the restored images.

### 2.3. Learning-Based Methods

Some learning-based methods are proposed to estimate the transmission maps t(x). For example, Cai et al. [17] and Ren et al. [32] first employ the CNNs for estimating the transmission map t(x) and use simple methods to calculate the atmospheric light *a* from the single hazy images. The paradigm of such models is shown in Figure 1b.

Although such CNN-based methods can remove haze by separately estimating the transmission map and the atmospheric light, it will introduce errors that affecting the image restoration. To avoid this problem, Zhang et al. [33]. adopted two CNN branches to estimate the atmospheric light and transmission map, respectively, and restoring clear images from the hazy images according to Equation (Equation 1). Compared with a separate estimation of atmospheric lights and transmission maps, the strategy proposed in [33] can significantly improve the dehazed results.

As shown in Figure 1c, several CNN-based algorithms regard image dehazing as enhancement tasks and directly recover clear images from hazy inputs. The GCANet [34] was proposed by Chen et al. for image dehazing with a new smoothed dilated convolution. The experimental results show that this method can achieve the better performances than previous representative dehazing methods. The training dataset mainly determines the performances of the algorithms. For example, when the image rain removal dataset replaces the training data, the algorithm can still achieve a good image rain removal performance as long as there is sufficient training. However, the pixel value of aerosols default to (255,255,255) in the traditional ASM and assume that the intensity of light in the scene is uniform.

The existing image dehazing methods and the traditional ASM give us the following inspirations:The image dehazing task can be viewed as a case of the decomposing images into clear layer and haze layer [35]. In the traditional ASM [27], the haze layer color is white by default, so many classical prior-based methods, such as [31], fail on white objects [2]. Therefore, it is necessary to improve the atmospheric model for adapting the different haze scenarios.The haze-free images obtained by Equation (Equation 4) has obvious defects when the value of atmospheric light received by the prior-based methods and the transmission maps obtained by the learning-based methods are used, due to they fail to cooperate with each other when two independent systems calculate two separate projects.The learning-based algorithms can directly output a clear images without using ASM. Such a strategy can achieve good dehazing performance on some datasets [36,37,38]. As CNN can have multiple outputs, one of the branches can directly output haze-free images.

As shown in Figure 1d, we propose another image dehazing paradigm for fire scenarios. In this paradigm, the deep learning model outputs four variables at the same time, and these four variables will act on the final dehazing result.

## 3. Proposed Method

To solve the problems encountered by the traditional image dehazing algorithms in the fire scenario, a novel network build with CNN and a new physical model are proposed in this paper. Unlike the general visual enhancement model, the proposed method is committed to adapting to the image degradation caused by different colors of the haze. Firstly, the proposed method adopts CNN similar to extract the low-dimensional features of the inputs and then outputs the scene transmittance map t(x), the atmospheric light value *a*, the color value of haze c(x), and the preliminary image recovered results $J_{Direct}$. After obtaining these crucial factors, CIANet adopts ESM to complete the fire scenario image dehazing task. Different from the traditional image dehazing methods, the proposed method can deal with the different scenes with different colors and degrees of haze and adapt to the overall atmospheric light value of the environment. This section will introduce the proposed CIANet in detail and explain how to use the ESM to restore the haze images in the fire scenario.

### 3.1. Color-Dense Illumination Adjustment Network

As described in [39], the hazy-to-transmission paradigms can achieve better performance than hazy-to-clear paradigms in uneven haze and changing illumination regions. Therefore, the proposed network utilizes these parameters to directly estimate clear images and ASM-related maps, i.e., illumination intensity, haze color, and transmission map. The proposed network is mainly composed of the following building blocks: (1) one shared encoder, which is constructed based on feature pyramid networks [40]; (2) three bottleneck block branches used to bifurcate features from the encoder to specific flows for decoders; (3) three separate decoders with different outputs. The complete network structure is shown in Figure 2.

Encoder: The structure of the shared encoder is shown in Table 1. DRHNet was initially being proposed for image dehazing and deraining, which proved that such an encoder could extract detailed features effectively by achieving a good performance in dehazing and deraining tasks. Therefore, the proposed CIANet utilizes the same model structure as the encoder part of DRHNET as the encoder.

Bottleneck: The bottleneck structure is used to connect the encoder and decoders. Ren et al. [32] prove that representing features at multiple scales is of great importance for image dehazing tasks. The Res2Net [41] represents multi-scale features that can expand the range of receptive fields for each layer. Gao et al. prove that the Res2Net can be plugged into the state-of-the-art CNN models, e.g., ResNet [42], ResNetXt [43] and DLA [44]. Due to the performance of the algorithms can be improved by increasing the receptive field of the convolution layer, Res2Net can significantly improve the receptive field of the CNN layer without incurring a significant increase in parameters. Different bottleneck structures connect to different decoders according to the function of the decoders. We use a shared bottleneck to estimate the global atmospheric light α and color *c*, which reducing the number of parameters in the network.

Decoders: The network includes three different decoders: the *t*-decoder, c&α-decoder and *J*-decoder, for predicting the color value *c*, global atmospheric light α, transmission map *t* and intermediate result *J*, respectively. The decoders share similar structures as the encoder but have different intermediate structures. In the c&α-decoder, we add a specially designed dilation inception module for the *J*-decoder, which we will describe in detail in the next section. Table 2 shows the details of the decoders.

### 3.2. Aerosol Scattering Model

The traditional ASM has been widely used in the image dehazing community, which can reasonably describe the imaging process in a hazy environment. However, many ASM-based dehazing algorithms suffer from the same limitation that may be invalid when the scene is inherently similar to the airlight [31]. The ineffectiveness is due to the assumption that the color of haze is white in the traditional ASM, which does not apply to all hazy environments due to the aerosols in the air may be mixed with some colored suspended particles. Therefore, the haze in different scenes has different color characteristics.

Due to the aerosol suspended in the air has a greater impact on the imaging results, we modified the traditional ASM and propose a new ESM. The default pixel value of haze in the air of traditional ASM is (255,255,255), which obviously does not conform to the appearance characteristics of degraded images in fire scenarios. In order to solve this problem, the ESM proposed in this paper combines color information c(x) to the haze and smoke generated in the fire scenarios. The schematic diagram of ESM is shown in Figure 3, and the formula expression is as follows:(3)I(x)=J(x)t(x)+α(1−t(x))c(x)
where, I(x) is the images captured by the devices, and J(x) is the clear images. Consistent with the traditional ASM, t(x) represents the transmission map, and α) represents the airlight value. The difference is that ESM introduces color information c(x), which is a 1∗3 array, including RGB values of haze color. In the model, Equation (Equation 3) is rewritten as follows:(4)$$J_{Final}=\frac{J_{Direct}-\alpha \left(c\left(x\right)-c\left(x\right)t\left(x\right)\right)}{t(x)}$$
where, $J_{Final}$ is the final output result, t(x) is the estimated transmission map, $J_{Direct}$ is the intermediate result produced by the proposed network directly, α is the global atmospheric light, and c(x) is the color.

### 3.3. Loss Function

#### 3.3.1. Mean Square Error

Recently, many data-driven image enhancement algorithms have used the mean square error (MSE) as the loss function to guide the direction of optimization [17,33]. To clearly describe the image pairs needed in calculating the loss function, let $J_{n}=\left(J_{n},n=1,\ldots ,N\right)$ represent the dehazing result of the proposed model, where $G_{n}=\left(G_{n},n=1,\ldots ,N\right)$ is the corresponding ground truth for the corresponding images. In the sequel, we omit the subscript *n* due to the inputs are independent. The mathematical expression of MSE is as follows:(5)L=‖G−J‖
where *G* is the ground truth images, *J* is the dehazed images.

#### 3.3.2. Feature Reconstruction Loss

We use both feature reconstruction loss [45] and MSE as the loss function. Li et al. prove that similar images are close to each other in their underlying and high-level features extracted from the deep learning model. This model is called “loss network” [45]. In this paper, we chose the VGG-16 model [46] as the loss network and used the reciprocal first, second, and third layers of the network as measurements to determine the loss function. The formula is as follows:(6)$$L_{p}=\overset{3}{\underset{i=1}{\sum }}\frac{1}{C_{i}H_{i}W_{i}}{∥VGG_{i}\left(R\right)-VGG_{i}\left(DRHNet\left(I\right)\right)∥}_{2}^{2}$$
where VGG() is the VGG-16 model, and *R* is the residual between the ground truth and the hazy images. *H*, *W*, and *C* represent the length, width, and the number of the feature map channels, respectively.

The final loss function can be described as follows:(7)$$L_{total}=L+\gamma L_{p}$$
where, γ is set to 0.5 in this paper, it should be noted that the design of loss function is not important in this paper, but the CIANet still can achieve good results with such simple loss function.

## 4. Experiment Result

We first introduce the experimental details in this section including the experimental datasets and the comparative algorithms, and then analyze and validate the effectiveness of different modules in the proposed CIANet. Finally, we compare with state-of-the-art dehazing methods by conducting extensive experiments both in synthetic and real-world datasets.

### 4.1. Experimental Settings

Network training setting: We adopt the same initialization scheme as DehazeNet [17] due to it is an effective dehazing algorithm based on the ASM and CNN. The weights of each layer are initialized by drawing randomly from a standard normal distribution, and the biases are set to 0. The initial learning rate is 0.1 and decreases by 0.01 every 40 epochs. The “Adam” optimization method [47] is adopted to optimize two networks.

The proposed network was trained end-to-end and implemented in the PyTorch platform, and all experiments were performed on a laptop with Intel(R) Core(TM) i7-8750H CPU @ 2.20GHz 2.20 GHz, 16GB RAM, and NVIDIA GeForce GTX 1070.

Dataset: Regarding the training data, 500 fire scenarios images with low image degradation were used as the training data. We uniformly sampled ten random gray values color∈[140,230] to generate the hazy images for each images. Therefore, a total of 5000 hazy images were generated for training. We named this dataset the realistic fire single image enhancement (RFSIE). Besides, the haze in the fire scenarios is usually non-homogeneous. Therefore, the training set provided in NTIRE’20 competition [48] can also be used as the training set for fire scenarios image dehazing algorithms. The images provided by NTIRE’20 were collected by a professional camera and haze generators to ensure the captured image pairs are the same except for the haze information. Moreover, due to the non-homogeneous haze was captured by [49], it has some similarities with the images of fire scenarios, so it is suitable for image enhancement for fire scenarios.

Compared methods: We compare our model with several state-of-the-art methods both on RFSIE and NTIRE’20, including He [31], Zhu [30], Ren [32], Cai [17], Li [2], Meng [49], Ma [50], Berman [29], Chen [34], Zhang [33] and Zheng [5].

### 4.2. Ablation Study

This section discusses different modules in CIANet and evaluates its impacts on the enhancement results.

Effects of ESM: Three groups of experiments were designed to verify the effectiveness of ESM. Table 3 presents the quantitative evaluation results of the proposed CIANet with different physical scattering models. In Table 3, $J_{Direct}$ represents the output of the *J-decoder*, and $J_{ASM}$ is the dehazing result obtained by using the traditional ASM with the default color of aerosol being white, $J_{ESM}$ represents the output of CIANet using the proposed ESM.

According to Table 3, $J_{ESM}$ achieves the best PSNR and SSIM values. The reasons are: (1) The image-to-image strategy usually disable to estimate the depth information of the image accurately, which leads to the inconspicuous dehazing performances in the area with dense haze. Therefore, the PSNR and SSIM values obtained by $J_{Direct}-only$ are slightly lower. (2) ESM can propose appropriate image restoration strategies for images with different styles and degrees of damage, while the traditional ASM assumes the color of aerosol is white by default, so it is easy to estimate the degree of damage falsely. Therefore, $J_{ASM}-only$ cannot achieve good dehazing performances. When the *T-decoder*, c&a-*decoder*, and *D-decoder* are trained together, the common backpropagation will promote each effect of the decoder and ensure that the encoder can extract the most effective haze features. Therefore, PSNR and SSIM values of $J_{Direct}$ and $J_{ASM}$ are slightly higher than those of $J_{Direct}-only$ and $J_{ASM}-only$.

Figure 4 shows the effectiveness of ESM intuitively. As can be seen from the first line of images in Figure 4, when the color of aerosol becomes milky white, the dehazing results of $J_{ASM}$ and $J_{ESM}$ are very similar. However, due to the obvious highlighted area in the fire scenarios, the traditional algorithm used to estimate the value of atmospheric light tends to overestimate the brightness, resulting in lower brightness of the dehazed result. When the color of aerosols in the air is dark, the dehazing effect of $J_{ESM}$ is better than that of $J_{ASM}$. The second row of images is from the position circled by the red rectangle. It can be seen that $J_{ASM}$ cannot restore the color of grass very well due to the default color of aerosol in ASM is lighter than the actual color. Hence, the performance of haze removal using ASM is lower than the real pixel value, and the overall color is dark.

### 4.3. Evaluation on Synthetic Images

We compare the proposed CIANet with some of the most advanced single-image dehazing methods on RFSIE, and adopt the indices of the peak signal-to-noise ratio (PSNR) and structural similarity index (SSIM) [51] to evaluate the quality of restored images.

Table 4 shows the quantitative evaluation results on our synthetic test dataset. Compared with other state-of-the-art baselines, the proposed CIANet can obtain higher PSNR and SSIM values. The average PSNR and SSIM of our dehazing algorithm are 18.37 db and 0.13 higher than the input hazy images, which indicates that the proposed algorithm can effectively remove haze and generate high-quality images.

As shown in Figure 5, the proposed method can generate clearer enhancement results than the most advanced image dehazing algorithms on hazy indoor images. The first and second rows of Figure 5 are the synthetic smoke and haze images of the indoor fire scenarios, and the third and fourth rows are the indoor haze images taken from the landmark dataset NTIRE2020. The dark channel algorithm proposed by He et al. produces some color distortion or brightness reduction (such as the first row and the third row of walls). The results show that the color attenuation prior algorithm proposed by Zhu et al. is not very effective, some images have a large amount of haze residues (such as the first row of images). The BCCR image dehazing algorithm proposed by Meng et al. can also cause image distortion or brightness reduction (such as the third and fourth rows of images). The Dehazenet algorithm proposed by Cai et al. can achieve a good dehazing effect in most cases, but there is an obvious haze residual in the first row of scene. The NLD algorithm proposed by Berman et al. can better complete the image dehazing task for indoor scenes, but there is less color distortion in the third row of images compared with the ground truth.

Figure 6 shows that the proposed method can also generate clearer images for outdoor scenes. Figure 6 can be divided into two parts. The first row and the second row of Figure 6 are taken from the landmark image dehazing database NTIRE2020, and the third and fourth rows of images are taken from the fire-related videos of traffic scenes captured by monitoring equipment. The image dehazing algorithm based on dark channel prior proposed by He et al. tends to estimate the transmission rate of the images and the atmospheric light value, resulting in large distortion in some parts of the images (such as the third and fourth rows of images). The color attenuation prior algorithm proposed by Zhu et al. can complete the image dehazing task to a certain extent, but there are still haze residues in the images (such as the second, third and fourth rows of images). The BCCR algorithm proposed by Meng et al. usually overestimates the brightness of the result. Although this estimation method can improve the detailed information of images, the result obtained is not similar to the ground truth. Dehazenet proposed by Cai et al. is based on the combination of deep learning and the traditional image dehazing algorithm. The results obtained by Dehazenet are very similar to that of CAP, and there are many haze residues. For outdoor images, the NLD algorithm can achieve good image dehazing effects, but some dehazing results are not as good as the results obtained with the proposed algorithm (such as the first row of images). Furthermore, the proposed algorithm can achieve better dehazing effects for outdoor scenes.

### 4.4. Evaluation on Real-World Images

Figure 7 shows the dehazing effect of the proposed algorithm compared with other state-of-the-art algorithms from real-world images. The first three rows of images are taken from NTIRE’20, and the last three rows are taken from the real images of fire scenarios. The most evident characteristic of the first three rows of images is that the thickness of the haze in image is uneven. For example, in the image of the second row in Figure 7, the haze thickness in the upper left corner of the image is obviously higher than that in the image area. The following three rows of images have similar characteristics with the first three rows of images, that is, different areas have varied degrees of damage. In addition, there is another remarkable feature on the last three rows of images, that is, each image has obvious highlighted area. These two characteristics basically cover all the features of the hazy images with the scenes of fire and smoke.

From the first row of images in Figure 7, we can see that the clarity of images obtained the proposed algorithm is significantly higher than that obtained with other algorithms, and there is no obvious artifact area. From the annotated area, the definition of this area is significantly higher than that of images obtained with other algorithms for comparison. On the second row of images in Figure 7, we can see that most algorithms cannot achieve the image dehazing well due to the left half of the images is seriously affected by haze. Compared with other algorithms, the proposed algorithm can still generate good performances. As shown in the marked area, the color saturation and clarity in the area can be reflected. The algorithm proposed in this paper can obtain better image dehazing effects. From the third row of images, we can still see that the algorithm can achieve better image dehazing effects. Both the brightness and the saturation of color are significantly higher than that of other algorithms.

In Figure 7, the images in the last three rows are obviously more complex than those in the first three rows. First of all, the images in the last three rows have obvious highlighted areas. Secondly, the haze color of the images in the last three rows is darker, which is more challenging than the images in the first three rows. It can be seen from the fifth row of Figure 7 that the algorithm proposed in this paper can remove most of the haze in the images, and basically maintain the structural information of the images, while other algorithms in comparison, such as AODNet, hardly remove any haze from the images. The image on the sixth row reflects that the proposed algorithm can almost remove all the haze in the images, and ensure that the result of the image will not change. In contrast, the dehazing effects of other algorithms are not obvious in this images, and it can be considered that the image dehazing task is completed to a large extent. Therefore, the CIANet proposed in this paper can be used to achieve dehazing of real hazy images to a certain extent.

### 4.5. Qualitative Visual Results on Challenging Images

Haze-free images: In order to prove the robustness of the proposed algorithm for all scenarios, we input the fire images which are not affected by the air particles into the network model. It can be seen from Figure 8 that the algorithm proposed in this paper has little effect on the fire scenarios image without fog, and it only slightly changes the color of the images, increasing the saturation of the image without damaging the structural information of the images. Therefore, this experiment proves the robustness of the algorithm. Hence, when the algorithm is embedded in the intelligent edge computing devices, it is not necessary to choose whether to run CIANet according to the change of situation.

### 4.6. Potential Applications

The CIANet proposed in this paper can effectively improve the visibility and clarity of the scene to promote the performance of other high-level visual tasks, which is the application significance of the algorithm proposed in this paper. To verify the proposed CIANet could benefit other vision tasks, we perform two applications: fire scenarios object detection and local keypoints matching. As can be seen from Figure 9 and Figure 10, the algorithm proposed in this paper can not only improve the visual quality and the quality of the input image, but also significantly improve the performance of subsequent important high-dimensional vision. The following two sections will discuss in detail the improvement of CIANet on object detection and local keypoint matching tasks.

#### 4.6.1. Object Detection

Most existing deep models for high-level vision tasks are trained using clear images. Such learned models will have low robustness when applied to degraded hazy fire scenarios images. In this case, the enhanced results can be useful for these high-level vision applications. To prove the proposed model can improve the detection precision, we analyze the performances of the object detection on our dehazed results. Figure 9 shows that using the proposed model as pre-processing can improve the detection performance.

#### 4.6.2. Local Keypoint Matching

We also adopt local keypoints matching, which aims to find correspondences between two similar scenarios, to test the effectiveness of the proposed CIANet. We utilize the SIFT operator for a pair or hazy fire scenarios images and as well as for the corresponding dehazed images. The matching result are shown in Figure 10. It is clear that the number of matched keypoints is significantly increased in the dehaze fire scenarios image pairs. This verifies that the proposed CIANet can recover the important features of the hazy images.

### 4.7. Runtime Analysis

The light-weight structure of CIANet leads to faster image enhancement. We select only one image from real-world and then repeat runing 100 times by different dehazing algorithms, on the same machine (Intel(R) Core(TM) i7-8750H CPU @2.20GHz and 16GB memory), without GPU acceleration. The per-image average running time of all models are shown in Table 5. Despite other slower MATLAB implementations, it is fair to compare DehazeNet (Pytorch version) and ours methods. The results illustrate the promising efficiency of CIANet.

## 5. Conclusions

This paper proposes CIANet, a color-dense illumination adjustment network that reconstructs clear fire scenario images via a novel ESM. We compare CIANet with the state-of-the-art dehazing methods, both on synthetic and real-world images both quantitatively (PSNR, SSIM) qualitatively (subjective measurements). The experimental results have shown that the superiority of the CIANet. Moreover, the experiments show that the proposed ESM is more reasonable than the traditional ASM in the fire scenarios imaging process. In the future, we will study the image enhancement algorithm of fire scenario on lidar image, so as to solve the problem that the traditional computer vision algorithm can not deal with the scene where there is a large amount of smoke and all visual information is lost.

## Figures and Tables

**Figure 1 sensors-22-00911-f001:**
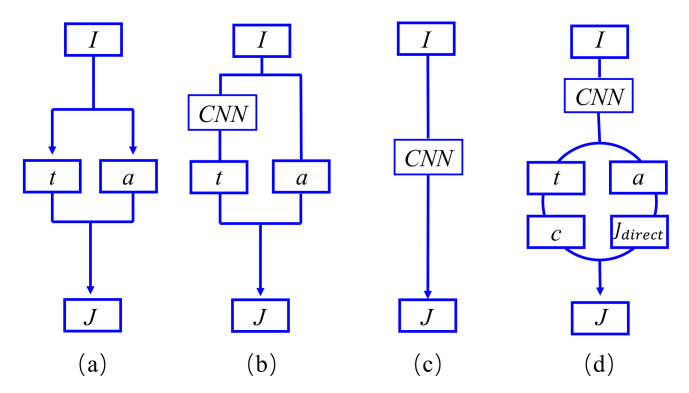
Different diagrams of dehazing schemes: (**a**) traditional two-step dehazing strategy; (**b**) estimate the transmission matrix through CNN; (**c**) end-to-end diagrams; (**d**) the proposed diagrams that estimates the transmission *t*, atmospheric light *a*, aerosol color *c*, and preliminary enhancement results $J_{direct}$.

**Figure 2 sensors-22-00911-f002:**
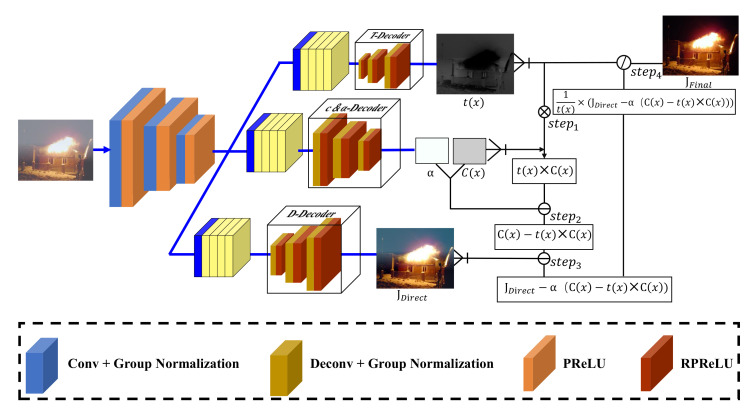
The structure of the proposed CIANet. All decoders are identical except for the *c&α decoder*, which outputs two floating point numbers The *T-decoder* and *D-decoder* outputs images.

**Figure 3 sensors-22-00911-f003:**
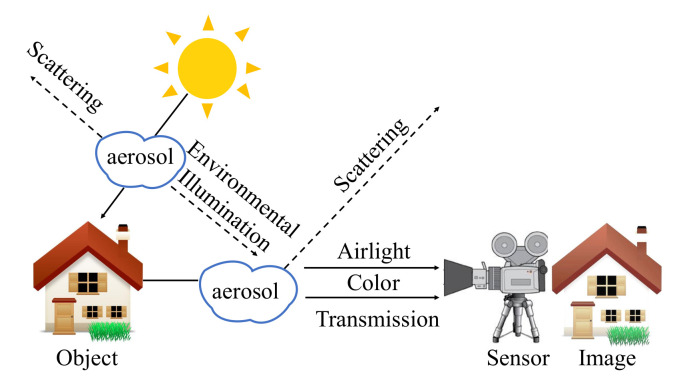
The imaging process in hazy fire scenarios. The transmission attenuation J(x)t(x) is caused by reducing reflection energy and making the color distortion and low brightness. The color value of aerosol in traditional ASM [27] is (255,255,255) by default, but the proposed ESM presents different visual characteristics of aerosols in different scenes.

**Figure 4 sensors-22-00911-f004:**
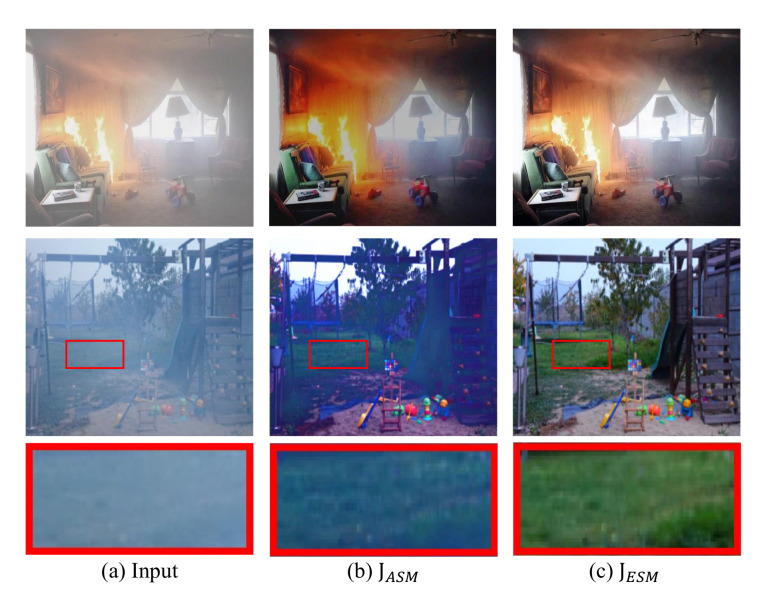
The effectiveness of the ESM on real-world images. The dehazed results of $J_{ESM}$ are much clearer than $J_{ASM}$.

**Figure 5 sensors-22-00911-f005:**
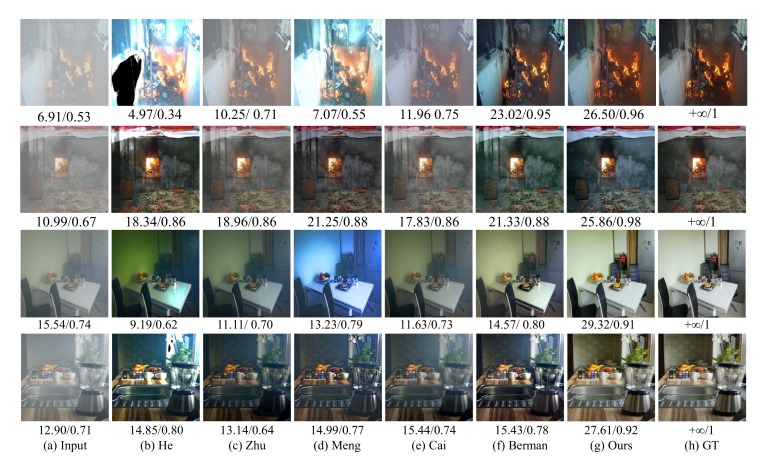
The performances of different image dehazing algorithms on synthetic indoor images.

**Figure 6 sensors-22-00911-f006:**
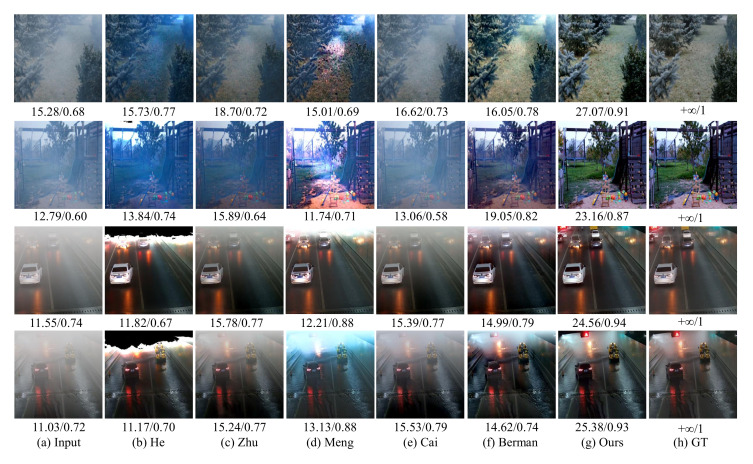
The performances of different image dehazing algorithms on synthetic outdoor images.

**Figure 7 sensors-22-00911-f007:**
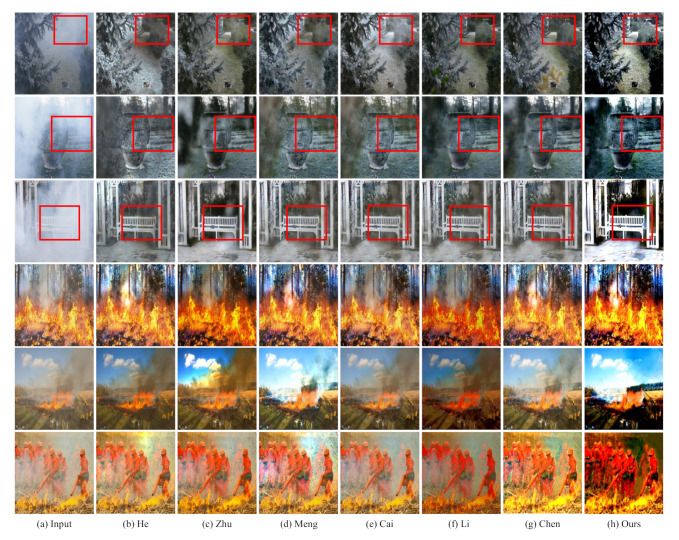
Visual comparisons on real-world images. The proposed method can effectively enhance the quality of different real-world hazy images with naturalness preservation.

**Figure 8 sensors-22-00911-f008:**
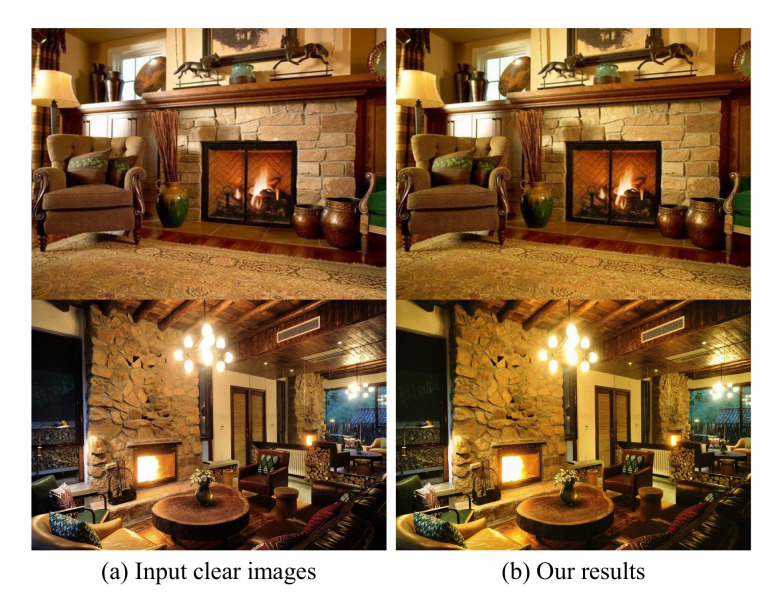
Examples for haze-free scenarios enhancement. (**a**): haze-free real photos with fire and. (**b**):enhancement results by CIANet.

**Figure 9 sensors-22-00911-f009:**
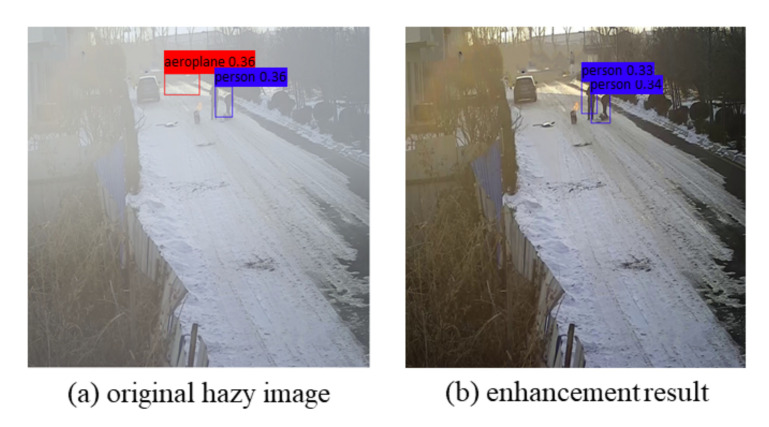
Pre-processing for object detection (Faster R-CNN [52], threshold = 0.3). (**a**): detection on hazy fire scenarios; (**b**): detection on the enhancement result.

**Figure 10 sensors-22-00911-f010:**
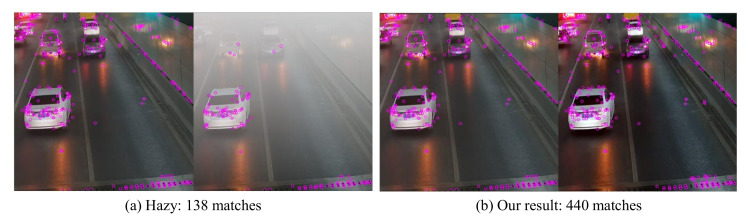
Local keypoints matching by applying the SIFT operator. Compared with the hazy images, the matching results shown that the proposed method can improved the quality of inputs significantly.

**Table 1 sensors-22-00911-t001:** Encoder structure.

	Enc. 1	Enc. 2	Enc. 3	Enc. 4	Enc. 5	Enc. 6	Enc. 7
Input	Input	Input	Input	Input	Input	Input	Input
Structure	$\left|\begin{array}{c} 3\times 3\;\mathrm{Conv}.\\ \mathrm{Stride}=2,\;\mathrm{Pool}=0 \end{array}\right|$	$\left|\begin{array}{c} 1\times 1\;\mathrm{Conv}\\ 3\times 3\;\mathrm{Conv} \end{array}\right|$	$\left|\begin{array}{c} 3\times 3\;\mathrm{Conv}\\ \mathrm{Stride}=2,\;\mathrm{Pool}=0 \end{array}\right|$	$\left|\begin{array}{c} 1\times 1\;\mathrm{Conv}\\ 3\times 3\;\mathrm{Conv} \end{array}\right|$	$\left|\begin{array}{c} 3\times 3\;\mathrm{Conv}\\ \mathrm{Stride}=2,\;\mathrm{Pool}=1 \end{array}\right|$	$\left|\begin{array}{c} 1\times 1\;\mathrm{Conv}\\ 3\times 3\;\mathrm{Conv} \end{array}\right|$	$\left|\begin{array}{c} 3\times 3\;\mathrm{Conv}\\ \mathrm{Stride}=2,\;\mathrm{Pool}=1 \end{array}\right|$
Output	310×230×32	310×230×32	155×115×64	155×115×64	78×58×128	78×58×128	39×29×256

**Table 2 sensors-22-00911-t002:** Decoder structure.

	Dec. 1	Dec. 2	Dec. 3	Dec. 4	Dec. 5	Dec. 6
	[Res. 1]	[Res. 2]	[Res. 3]	[Res. 4]	[Res. 5]	[Res. 6]
T-Decoder	$\left[\;\;\begin{array}{c} 1\times 1\;\mathrm{Conv},\;256\\ 3\times 3\;\mathrm{Conv},\;256\\ 1\times 1\;\mathrm{Conv},\;512 \end{array}\;\;\right]$	$\left[\;\;\begin{array}{c} 1\times 1\;\mathrm{Conv},\;512\\ \;\mathrm{upsample}\;2 \end{array}\;\;\right]$	$\left[\;\;\begin{array}{c} 1\times 1\;\mathrm{Conv},\;256\\ 3\times 3\;\mathrm{Conv},\;256\\ 1\times 1\;\mathrm{Conv},\;512 \end{array}\;\;\right]$	$\left[\;\;\begin{array}{c} 1\times 1\;\mathrm{Conv},\;512\\ \;\mathrm{upsample}\;2 \end{array}\;\;\right]$	$\left[\;\;\begin{array}{c} 1\times 1\;\mathrm{Conv},\;256\\ 3\times 3\;\mathrm{Conv},\;256\\ 1\times 1\;\mathrm{Conv},\;512 \end{array}\;\;\right]$	$\left[\;\;\begin{array}{c} 1\times 1\;\mathrm{Conv},\;512\\ \;\mathrm{upsample}\;2 \end{array}\;\;\right]$
	39×29×512	78×58×512	78 ×58×512	156×116×512	155×115×512	310×230×512
	Dec. 1	Dec. 2	Dec. 3	Dec. 4	Dec. 5	Dec. 6
	[Res, 4, Trans. 2]	[Res. 4, Trans. 2]	[Res. 4, Trans. 2]	[Res. 4, Trans. 2]	[Res. 4, Trans. 2]	[Res. 4, Trans. 2]
c&a-Decoder	$\left[\;\;\begin{array}{c} 1\times 1\;\mathrm{Conv},\;256\\ 3\times 3\;\mathrm{Conv},\;256\\ 1\times 1\;\mathrm{Conv},\;512 \end{array}\;\;\right]$	$\left[\;\;\begin{array}{c} 1\times 1\;\mathrm{Conv},\;256\\ \;\mathrm{downsample}\;2 \end{array}\;\;\right]$	$\left[\;\;\begin{array}{c} 1\times 1\;\mathrm{Conv},\;256\\ 3\times 3\;\mathrm{Conv},\;256\\ 1\times 1\;\mathrm{Conv},\;512 \end{array}\;\;\right]$	$\left[\;\;\begin{array}{c} 1\times 1\;\mathrm{Conv},\;256\\ \;\mathrm{downsample}\;2 \end{array}\;\;\right]$	$\left[\;\;\begin{array}{c} 1\times 1\;\mathrm{Conv},\;256\\ 3\times 3\;\mathrm{Conv},\;256\\ 1\times 1\;\mathrm{Conv},\;512 \end{array}\;\;\right]$	$\left[\;\;\begin{array}{c} 1\times 1\;\mathrm{Conv},\;256\\ 3\times 3\;\mathrm{Conv},\;256\\ 1\times 1\;\mathrm{Conv},\;512 \end{array}\;\;\right]$
	39×29×512	20×14×512	20×14×512	10×7×512	10×7×512	10×7×512
	Dec. 1	Dec. 2	Dec. 3	Dec. 4	Dec. 5	Dec. 6
	[Res, 4, Trans. 2]	[Res. 4, Trans. 2]	[Res. 4, Trans. 2]	[Res. 4, Trans. 2]	[Res. 4, Trans. 2]	[Res. 4, Trans. 2]
J-Decoder	$\left[\;\;\begin{array}{c} 1\times 1\;\mathrm{Conv},\;256\\ 3\times 3\;\mathrm{Conv},\;256\\ 1\times 1\;\mathrm{Conv},\;512 \end{array}\;\;\right]$	$\left[\;\;\begin{array}{c} 1\times 1\;\mathrm{Conv},\;256\\ \;\mathrm{downsample}\;2 \end{array}\;\;\right]$	$\left[\;\;\begin{array}{c} 1\times 1\;\mathrm{Conv},\;256\\ 3\times 3\;\mathrm{Conv},\;256\\ 1\times 1\;\mathrm{Conv},\;512 \end{array}\;\;\right]$	$\left[\;\;\begin{array}{c} 1\times 1\;\mathrm{Conv},\;256\\ \;\mathrm{downsample}\;2 \end{array}\;\;\right]$	$\left[\;\;\begin{array}{c} 1\times 1\;\mathrm{Conv},\;256\\ 3\times 3\;\mathrm{Conv},\;256\\ 1\times 1\;\mathrm{Conv},\;512 \end{array}\;\;\right]$	$\left[\;\;\begin{array}{c} 1\times 1\;\mathrm{Conv},\;512\\ \;\mathrm{upsample}\;2 \end{array}\;\;\right]$
	39×29×512	78×58×512	78×58×512	156×116×512	155×115×512	310×230×512

**Table 3 sensors-22-00911-t003:** Average PSNR (dB) and SSIM results of different outputs from CIANet on RFSIE and NTIRE’20. The first and second best results are highlighted in red and blue.

		NTIRE’20		RFSIE		Time/Epoch
Metric		PSNR	SSIM	PSNR	SSIM	
CIANet	J${}_{direct}$	13.11	0.56	24.81	0.82	63 min
	$J_{ASM}$	14.23	0.58	25.34	0.81	
	$J_{ESM}$	18.34	0.62	31.22	0.91	
$J_{direct}$-only		12.11	0.51	24.96	0.78	21 min
$J_{ASM}$-only		14.21	0.59	25.91	0.80	21 min

**Table 4 sensors-22-00911-t004:** Average PSNR and SSIM results on RFSIE. The first, second, and third best results are highlighted in red, blue, and bold, respectively.

Methods	Hazy	He	Zhu	Ren	Cai	Li	Meng	Ma	Berman	Chen	Zhang	Zheng	Ours
PSNR	12.85	17.42	19.67	23.68	21.95	23.92	24.94	26.19	26.95	**27.25**	27.36	26.11	31.22
SSIM	0.78	0.80	0.82	**0.85**	0.87	0.82	0.82	0.82	**0.85**	**0.85**	**0.85**	0.82	0.91

**Table 5 sensors-22-00911-t005:** Comparison of average model running time (in seconds).

Image Size	480×640	Platform
He	26.03	Matlab
Berman	8.43	Matlab
Meng	2.19	Matlab
Ren	2.01	Matlab
Zhu	1.02	Matlab
Cai (Matlab)	2.09	Matlab
Cai (Pytorch)	6.31	Pytorch
CIANet	4.77	Pytorch

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
