# Peer review of "Color-Dense Illumination Adjustment Network for Removing Haze and Smoke from Fire Scenario Images"

_sensors, 2022, doi:10.3390/s22030911_

Round 1

Reviewer 1 Report

This study present color-dense illumination adjustment network (CIANet) for joint recovery of transmission matrix, illumination intensity, and the dominant color of aerosols from a single image. There are some points should be solved in the revised version.

Comment: Why is it important to remove Haze and Smoke from Fire Scenario Images? What and where are its applications?

Comment: “Recently, many dehazing algorithms for a single images have been proposed,” it should cite some related work in order to sustain the statement.

Comment: “This paper modifies ASM and proposes a new imaging model named ESM for enhancing the quality of images or videos captured from fire scenarios”. What is novelty claimed by authors?

Comment: “When concatenating CIANet with Faster R-CNN, we witness an improvement of the objection performance with a large margin.” Could the authors explain better what they mean and how they did it?

Comment: How to optimize hyperparameters during training?

Comment: There should be a separate section “4.Discussion” before “5.Conclusion” and authors should discuss the limitations of their work at the end of Discussion section.

Author Response

    Dear reviewer,

Thank you for your letter and for the comments concerning our manuscript entitled “Color-dense Illumination Adjustment Network for Removing Haze and Smoke from Fire Scenario Images” (sensors-1547311), as well as the important guiding significance to our researches. We have studied comments carefully and have made correction which we hope meet with approval. Revised portion are displayed in this file. 

    Yours sincerely

Chuansheng Wang

Reviewer 2 Report

The authors propose a novel learning-based dehazing model to improve the quality of images captured from fire scenarios, built with CNN and a physical imaging model.
To improve the effect of image dehazing, they improve the existing atmospheric scattering model (ASM) and propose a new ASM called the aerosol scattering model (ESM).

The topic is very interesting.

The authors describe very clearly the challenges in the field of image dehazing, the related works and the proposed method.

The experimental results are well described and validate the proposed approach.

Moreover, a comparison with the state-of-the art methods has been conducted to demonstrate the improvement achieved with the proposed method.

For there reasons, I think that this work is fully suitable for publication.

Minor comments:
- even if used in the abstract, acronyms must be specified the first time they are used in the body of the paper (e.g. atmospheric scattering model (ASM) at line 47; aerosol scattering model (ESM) at line 52).

Author Response

Dear Reviewer:

Thank you for your letter and for the comments concerning our manuscript entitled “Color-dense Illumination Adjustment Network for Removing Haze and Smoke from Fire Scenario Images” (sensors-1547311), as well as the important guiding significance to our researches. We have studied comments carefully and have made correction which we hope meet with approval. Revised portion are displayed in this file.

Best regards,
Chuansheng Wang

Reviewer 3 Report

The article is very well written. The method is clear and the literature related to the problem has been reviewed. The authors carried out extensive experiments including a complete comparison with methods reported in the state of the art, both in internal and external environments, as well as in synthetic and real images. They also added possible extensions to the method so that it can be used in conjunction with other algorithms (SIFT). I think the article is very well done and I see no shortcomings.

Minor
Page 3. Eq1. A --> α
Page 10. Fig5. 1 infoor --> indoor

Author Response

(The authors gave the same response as above.)

Round 2

Reviewer 1 Report

The authors have answered my questions satisfactorily.